# The 6-Month Antibody Durability of Heterologous Convidecia Plus CoronaVac and Homologous CoronaVac Immunizations in People Aged 18–59 Years and over 60 Years Based on Two Randomized Controlled Trials in China

**DOI:** 10.3390/vaccines11121815

**Published:** 2023-12-04

**Authors:** Hudachuan Jiang, Pengfei Jin, Xiling Guo, Jiahong Zhu, Xue Wang, Peng Wan, Jingxuan Wan, Jingxian Liu, Jingxin Li, Fengcai Zhu

**Affiliations:** 1School of Public Health, Southeast University, 87 Dingjiaqiao Avenue, Nanjing 210009, China; 15951884741@163.com; 2NHC Key Laboratory of Enteric Pathogenic Microbiology, Jiangsu Provincial Center for Disease Control and Prevention, 172 Jiangsu Avenue, Nanjing 210009, China; kingpfph@sina.com (P.J.); guoxlnj@jscdc.cn (X.G.); oliveppl@163.com (J.L.); 3School of Science, China Pharmaceutical University, Nanjing 210009, China; 4Lianshui County Center for Disease Control and Prevention, Huaian 223400, China; lscdczjh@126.com; 5CanSino Biologics Inc., Tianjin 300450, China; xue.wang@cansinotech.com (X.W.); peng.wan@cansinotech.com (P.W.); jingxuan.wan@cansinotech.com (J.W.); 6School of Public Health, National Vaccine Innovation Platform, Nanjing Medical University, Nanjing 210009, China

**Keywords:** COVID-19 vaccine, heterologous immunization, prime-boost, antibody durability

## Abstract

Previous reports have shown that heterologous boosting with the AD5-vectored COVID-19 vaccine Convidecia based on a primary series of two doses of inactivated vaccine induces increasing immune responses. However, the immune persistence until 6 months after the heterologous prime-boost immunization was limited. Participants were from two single-center, randomized, controlled, observer-blinded trials, which involved individuals of 18–59 years of age and over 60 years of age. Eligible participants who previously primed with one dose or two doses of CoronaVac were stratified and randomly assigned to inoculate a booster dose of Convidecia or CoronaVac. Neutralizing antibodies against a live SARS-CoV-2 prototype virus and Delta and Omicron (B.1.1.529) variants, pseudovirus neutralizing antibodies against Omicron BA.4/5 variants, and anti-SARS-CoV-2 RBD antibodies at month 6 were detected, and the fold decreases and rate difference were calculated by comparing the levels of antibodies at month 6 with the peak levels at month 1. The neutralizing antibody titers against prototype SARS-CoV-2, RBD-specific IgG antibodies, and the Delta variant in the heterologous regimen of the CoronaVac plus Convidecia groups were significantly higher than those of the homologous prime-boost groups. In three-dose regimen groups, the geometric mean titers (GMTs) of neutralizing antibodies against prototype SARS-CoV-2 were 30.6 (95% CI: 25.1; 37.2) in the heterologous boosting group versus 6.9 (95% CI: 5.6; 8.6) in the homologous boosting group (*p* < 0.001) at month 6 in participants aged 18–59 years, and in the two-dose regimen, the neutralizing antibody GMTs were 8.5 (95% CI: 6.2; 11.7) and 2.7 (2.3 to 3.1) (heterologous regimen group versus CoronaVac regimen group, *p* < 0.001). Participants aged over 60 years had similar levels of neutralizing antibodies against the prototype, with GMTs of 49.1 (38.0 to 63.6) in the group receiving two doses of CoronaVac plus one dose of Convidecia versus 9.4 (7.7 to 11.4) in the group receiving three doses of CoronaVac (*p* < 0.001) and 11.6 (8.4 to 16.0) in the group receiving one dose of CoronaVac and one dose of Convidecia versus 3.3 (2.7 to 4.0) in the group receiving two doses of CoronaVac (*p* < 0.001). Compared with day 14, over sixfold decreases in neutralizing antibody GMTs were observed in the heterologous groups of the three- or two-dose regimen groups of younger and elderly participants, while in the homologous regimen groups, the GMTs of neutralizing antibodies decreased about fivefold in the two age groups. The heterologous prime-boost regimen with two doses of CoronaVac and one dose of Convidecia was persistently more immunogenic than the regimen of the homologous prime-boost with three doses of CoronaVac.

## 1. Introduction

CoronaVac (COVID-19 vaccine, inactivated, developed by SINOVAC) has been widely used as the primary series of COVID-19 vaccination in China, Brazil, Mexico, Pakistan, Chile, Egypt, Indonesia, Nepal, Turkey, etc., to prevent the SARS-CoV-2 pandemic worldwide [1]. However, significant waning of the protective immunity produced by the inactivated vaccine has occurred as time has passed, likely amplified by the prevalence of Variants of Concern (VOCs) [2,3,4,5].

Mutations occur in the receptor-binding domain (RBD) of VOCs, especially the Omicron variants of BA.4 and BA.5, which restricts the neutralization of prototype-based SARS-CoV-2 vaccines [6,7,8,9]. Only half of vaccinees who received mRNA vaccines, adenovirus-vectored vaccines, or inactivated vaccines had detectable neutralizing antibodies against the Omicron variant 3–6 months after the completion of primary immunization, but an additional heterologous or homologous booster increased the breadth and cross-reactivity of humoral immunity against the Omicron variant [10,11,12,13]. Therefore, countries that employed inactivated vaccines for primary immunization are suggested to consider vector-based or mRNA COVID-19 vaccines for additional doses on the basis of the interim recommendations for heterologous COVID-19 vaccine schedules by the World Health Organization (WHO) [14].

Convidecia is a recombinant adenovirus type-5 COVID-19 vaccine that was developed by CanSinoBIO and has been authorized for use in China, Chile, Mexico, Pakistan, Malaysia, Indonesia, etc. [1], and granted an emergency-use listing by the WHO [15]. In February 2022, China approved a heterologous prime-boost immunization schedule for COVID-19 vaccines for people aged 18 years or older. Those who have received two doses of inactivated vaccines are encouraged to receive a booster dose of the Ad5-vectored vaccine Convidecia [16].

Now, available reports about the long-term immunogenicity of heterologous prime-boost immunization are limited, and a few studies have shown the specific neutralizing antibodies, binding antibodies, and T-cell immune responses persisting for 2–6 months after the booster dose [17,18]. The durability of the immune responses elicited by heterologous prime-boost immunization schedules is in need of assessment, particularly for the Omicron variant.

In our previous studies, we have shown that heterologous boosting with Convidecia induced a stronger increase in neutralizing antibodies than homologous boosting with CoronaVac in a healthy population who inoculated CoronaVac [19]. Briefly, the geometric mean titers (GMTs) of neutralizing antibodies against wild-type SARS-CoV-2 and the Delta variant in participants boosting with Convidecia at day 14 were 5.9-fold and 6.8-fold higher than those in participants boosting with CoronaVac. Similar antibody results were observed in people aged over 60 years [20].

This study reports the immune persistence of a heterologous prime-boost of CoronaVac plus Convidecia in two phase 4 trials, with adults aged 18–59 years and elderly adults aged 60 years or older, to evaluate antibody durability up to 6 months after a booster vaccination.

## 2. Methods

### 2.1. Study Design and Participants

We conducted two single-center, randomized, controlled, observer-blinded trials, one investigating participants aged 18–59 years and the other investigating participants aged over 60 years old in Lianshui County, Jiangsu Province (NCT04892459, NCT04952727). Both studies aimed to assess the safety and immunogenicity of heterologous prime-boost immunizations of Convidecia following the primary series of CoronaVac. As previously described [19], healthy participants at the ages of 18–59 years and 60 years or older who had received the one-dose schedule of the inactivated SARS-CoV-2 vaccine CoronaVac in the past 1–3 months or the two-dose schedule of CoronaVac in the past 3–6 months were enrolled (Table 1). Body temperature and blood pressure were tested, and nasopharyngeal swabs and medical histories were collected. Participants who were clinically confirmed or laboratory confirmed to have COVID-19 or SARS-CoV-2 infection were excluded. Other exclusion criteria included the history of clinical or virologic SARS-CoV-2 infection or COVID-19, convulsion, serious acute hypersensitive reaction to vaccines, acute febrile diseases or infectious diseases, asplenia or functional asplenia, any serious chronic conditions or urticaria within 1 year, and those who had received anti-tuberculosis treatment, immunosuppressive therapy, or anti-allergy therapy, known infection with human immunodeficiency virus, women with positive urine pregnancy tests, and so on.

### 2.2. Randomization

A stratified interactive web-based response randomization system was used according to the number of priming doses that the participants had received. In addition, eligible participants who had completed the two-dose primary series of CoronaVac were randomly assigned at a ratio of 1:1 to receive a booster dose of Convidecia (group A) or CoronaVac (group B), while others who had primed with the one-dose of CoronaVac were randomized at a ratio of 1:1 to receive another dose of Convidecia (group C) or CoronaVac (group D) (Figure 1). Randomization lists were generated by an independent statistician using SAS (version 9.4).

We blinded investigators, laboratory staff, and outcome assessors to the allocation of treatment groups but not to the three-dose or two-dose schedule. As the vials and syringes for Convidecia and CoronaVac were different, unblinded personnels were in charge of vaccine preparation and administration.

### 2.3. Outcomes

The primary endpoint for safety was the incidence of adverse reactions within 28 days after vaccination and the primary endpoint for immunogenicity was the GMTs of neutralizing antibodies against a live SARS-CoV-2 prototype virus at day 14 after the booster dose, both of which have been previously reported [19]. Here, we show the neutralizing antibodies against the live SARS-CoV-2 prototype virus, the Delta variant, and the Omicron variant, pseudovirus neutralizing antibodies against the Omicron BA.4/BA.5 variant, and anti-SARS-CoV-2 receptor-binding- domain (RBD) antibodies up to 6 months after the boost vaccination.

### 2.4. Serologic Assays

An amount of 20 mL of venous blood per person was collected with an anticoagulant collection vessel and was stored at −20 °C. Live-virus neutralizing antibody titers were measured by using a cytopathic effect-based microneutralization assay with the prototype SARS-CoV-2 virus isolate Beta CoV/Jiangsu/JS02/2020 (GISAID EPI_ISL_411952), Delta hCoV-19/China/JS07/2021 (GISAID EPI_ISL_4515846), and Omicron isolate hCoV-19/Jiangsu/ JS01/2022 (GISAID EPI_ISL_12511653) variants. Serum dilution for the microneutralization assay started from 1:4 and then mixed with the same volume of virus to achieve a 50% infectious dose of 100 per well. The reported titer was the reciprocal of the highest dilution that protected ≥50% of cells from a cytopathic effect under an inverted microscope. Seropositivity of neutralizing antibody was defined as titer ≥ 1:4. A pseudovirus-based neutralization assay was conducted to detect neutralizing titers against BA.4/BA.5 variants with a detectable antibody titer ≥ 1:30. Pseudotyped viruses were produced by ACE2-293T cells (Vazyme Biotech Co., Ltd., Nanjing, China), transfected with S protein-expressing plasmids of Omicron BA.4/BA.5 and infected with an HIV pseudovirus system [21]. Anti-SARS-CoV-2 RBD-specific antibody response was measured using an indirect ELISA assay by the commercial Anti-SARS-CoV-2 RBD IgG ELISA kit (Vazyme Biotech Co., Ltd., Nanjing, China) with a cut-off titer of 1:10.

### 2.5. Sample Size

The sample size was based on the hypothesis that neutralizing antibodies elicited by a heterologous booster dose would be not inferior to those of the homologous booster dose (group A vs. group B). The baseline GMT of neutralizing antibodies was assumed to be approximately 1:40 before the booster, while 1:80 was expected after one dose of inactivated vaccine and 1:160 was estimated after receiving one dose of Convidecia. The standard deviation of the GMTs for the two groups was estimated to be 4. A sample size of at least 86 per group provides more than 90% power to identify the log-transformed neutralization titers in group A as superior to those in group B. In order to meet both assumptions and consider a drop off of 10%, the sample size was designed to be about 100 people per group. Considering the difficulty of recruiting, 50 persons were allocated to both group C and group D for exploratory purposes. The total sample size was about 300.

### 2.6. Statistical Analysis

The serum neutralizing antibody GMTs and RBD-specific antibodies in serum were calculated with two-sided 95% confidence intervals. A *χ*^2^ test or Fisher’s exact test was used to analyze categorical data, the *t*-test was used for log-transformed antibody titers, and the Wilcoxon rank-sum test was used for data not following a normal distribution. The above-mentioned analyses were performed in the intervention-modified per protocol set, including all the participants who received the vaccinations and finished serum collection according to the protocol. Statistical analyses were performed using SAS (version 9.4) or GraphPad Prism 8.0.1.

## 3. Results

### 3.1. Study Participants

A total of 280 participants aged 18–59 years (90 in group A, 95 in group B, 47 in group C, and 48 in group D, Figure 1A) and 259 participants aged 60 years or older completed blood collection (84 in group A, 86 in group B, 45 in group C, and 44 in group D, Figure 1B) at month 6. In people aged 18–59 years, the median time interval between the second dose and the booster dose were 3.2 months in group A, 3.8 months in group B, and 1.8 months in the two-dose regimen cohorts. In people aged over 60 years, time since the last priming dose of inactivated vaccine was 4.9 months in the three-dose regimen cohorts and 1.1 months in the two-dose regimen cohorts. More detailed demographic characteristics are shown in Table 1.

### 3.2. The Neutralizing Antibodies against SARS-CoV-2 Prototype and RBD-Specific IgG Antibodies

#### 3.2.1. The Neutralizing Antibodies against SARS-CoV-2 Prototype Antibodies

As previously reported, the GMTs of neutralizing antibodies against the SARS-CoV-2 prototype in the 18–59 age group significantly increased post-vaccination and reached to a peak level of 150.3 at day 14, and participants aged over 60 years had similar neutralizing antibody levels as those of the 18–59 age group (Appendix A) [19,20]. In addition, heterologous boosting with Convidecia induced significantly increased neutralizing antibody levels compared to homologous boosting with CoronaVac. At month 6, in participants aged 18–59 years receiving the three-dose regimen cohorts, the GMT of neutralizing antibodies against the SARS-CoV-2 prototype was 30.6 (95% CI: 25.1 to 37.2) in the heterologous boosting group, which was significantly higher than that (6.9 [95% CI: 5.6 to 8.6]) in the homologous boosting group (*p* < 0.001). Neutralizing antibody GMTs were 8.5 (95% CI: 6.2 to 11.7) in adults receiving one dose of CoronaVac plus Convidecia versus 2.7 (95% CI: 2.3 to 3.1) in the group receiving two doses of CoronaVac (*p* < 0.001). For the elderly, the neutralizing antibody GMTs against the SARS-CoV-2 prototype were more robust in the Convidecia boosting group versus the CoronaVac boosting group, with GMTs of 49.1 (95% CI: 38.0 to 63.6) and 9.4 (95% CI: 7.7 to 11.4) in the three-dose regimen groups (*p* < 0.001) and GMTs of 11.6 (95% CI: 8.4 to 16.0) and 3.3 (95% CI: 2.7 to 4.0) in the two-dose regimen groups (*p* < 0.001) (Figure 2A, Appendix A). Compared with the peak neutralizing antibodies at day 14, the GMTs of neutralizing antibodies decreased by 6.5-fold in the heterologous boosting group and 4.9-fold in the homologous boosting group in three-dose regimen groups at month 6. In the meantime, the GMTs of neutralizing antibodies decreased by 6.4-fold and 4.8-fold in the two-dose regimen groups for participants aged 18–59 years. For participants aged over 60 years, neutralizing antibodies decreased by 6.1-fold and 5.1-fold at month 6 in comparison to the level at day 14 in the heterologous boosting group and the homologous boosting group for the three-dose regimen cohort and by 6.5-fold and 3.6-fold in the heterologous group and the homologous group, respectively, for the two-dose regimen cohort (Appendix A).

Likewise, the proportion of participants with seropositivity of neutralizing antibodies in the heterologous groups was higher than that in the homologous groups in adults aged 18–59 years at month 6 (96.7% (95% CI, 90.6% to 99.3%) in the heterologous boosting group vs. 72.6% (95% CI: 62.5% to 81.3%) in the homologous boosting group, 78.7% (95% CI: 64.3% to 89.3%) in the heterologous group vs. 27.1% (95% CI: 15.3% to 41.8%) in the homologous group, *p* values < 0.001). Similar seropositivity rates were found in the elderly, and the proportions of participants with seropositivity were higher in the heterologous boosting group (100.0% (95% CI: 95.7% to 100.0%)) and the heterologous group (91.1% (95% CI: 78.8% to 97.5%)) than in the homologous boosting group (93.0% (95% CI: 85.4% to 97.4%)) and the homologous group (40.9% (95% CI: 26.3% to 56.8%)) (*p* = 0.013, *p* < 0.001) (Appendix A).

#### 3.2.2. The RBD-Specific IgG Antibodies

As for RBD-specific IgG antibody levels, the GMTs were 154.1–3090.3 in adults aged 18–59 years and 146.5–3180.5 in the elderly group at day 14 (Appendix A). At month 6, IgG GMTs observed in the heterologous groups for participants aged 18–59 years were 270.0 (95% CI: 226.3 to 322.1) in the three-dose regimen group and 41.7 (95% CI: 29.9 to 58.1) in the two-dose regimen group. These results were significantly higher than those in the homologous groups: 60.5 (95% CI: 48.6 to 75.4) in the three-dose regimen group and 14.6 (95% CI: 11.5 to 18.5) in the two-dose regimen group. Elderly people also demonstrated stronger RBD-specific IgG levels for receiving heterologous doses with GMTs that were 49.1 (95% CI: 38.0 to 63.6) in the Convidecia boosting group and 9.4 (95% CI: 7.7 to 11.4) in the CoronaVac boosting group (*p* < 0.001) and 11.6 (95% CI: 8.4 to 16.0) in the Convidecia group and 3.3 (95% CI: 2.7 to 4.0) in the CoronaVac group (*p* < 0.001) (Figure 2B, Appendix A). RBD-specific IgG antibody levels decreased more evidently at month 6, with GMTs 6.1- to 22.6-fold lower in the 18–59 age group and 8.4- to 24.5-fold lower in the elderly group (Appendix A).

Although significant decreases in RBD-specific IgG antibody levels were observed, the proportions of participants with seropositive antibodies were still high at month 6. In the three-dose regimen groups, seropositivity was 100% in the 18–59 age group, and for two-dose regimen groups, participants vaccinated with two doses of CoronaVac showed the lowest seropositivity rate of 85.4%. The proportions of participants with seropositivity were 75.0–100% in the elderly, which were equivalent to those for young people (Appendix A).

### 3.3. The Neutralizing Antibodies against the Delta and Omicron (B.1.1.529) Variants

#### 3.3.1. The Neutralizing Antibodies against the Delta Variants

Our previous studies have reported the neutralizing antibody levels against the Delta variant at day 14 for people aged 18–59 years and over 60 years old [19,20]. At month 6, participants in the heterologous groups had a significant higher seropositivity rate of neutralizing antibodies against Delta than that of the homologous groups for participants aged 18–59 years (96.7% (95% CI: 90.6% to 99.3%) vs. 85.3% (95% CI: 76.5% to 91.7%) in the two three-dose regimen groups, *p* = 0.007; 76.6% (95% CI: 62.0% to 87.7%) vs. 52.1% (95% CI: 37.2% to 66.7%) in the two two-dose regimen groups, *p* = 0.013). Similarly, for participants aged over 60 years or older, participants boosted with Convidecia (64.3% (95% CI: 53.1% to 74.4%) had reportedly higher seropositivity rates than those boosted with CoronaVac (26.6% (95% CI: 17.8% to 37.4%) (*p* < 0.001)), and for the two-dose regimen cohorts, the percentage of participants with seropositivity was 11.1% (95% CI: 3.7% to 24.1%) in the heterologous group, also significantly higher than that of the homologous group (4.5% (95% CI: 0.6% to 15.5%) (*p* < 0.001) (Appendix A).

Neutralizing antibody GMTs of 11.7 (95% CI: 9.6 to 14.1) and 5.2 (95% CI: 4.5 to 5.9) were noted in the heterologous boosting group and the homologous boosting group of the three-dose groups, and values of 4.8 (95% CI: 3.8 to 6.1) and 3.0 (95% CI: 2.6 to 3.3) were noted in the heterologous group and the homologous group of the two-dose groups at month 6. Participants aged over 60 years in the Convidecia boosting group (GMT: 9.2 (95% CI: 6.9 to 12.3)) had higher antibody levels than the CoronaVac boosting group (GMT: 2.7 (95% CI: 2.4 to 3.0)) (Figure 3A, Appendix A).

In the Convidecia boosting group, participants had the most significant decreases in neutralizing antibodies against Delta at month 6 compared to day 14, with more than fourfold declines, whether for people aged 18–59 years or for elderly participants (Appendix A).

#### 3.3.2. The Neutralizing Antibodies against the Omicron (B.1.1.529) Variants

In the 18–59 age cohort, only participants receiving the three-dose regimen had positive neutralizing antibodies against Omicron (B.1.1.529) at day 14, with seropositivity rates of 90.0% (95% CI: 73.5% to 97.9%) and 53.3% (95% CI: 34.3% to 71.7%) in the heterologous boosting group and the homologous boosting group, respectively. The neutralizing antibody levels against Omicron (B.1.1.529) for participants over 60 years at day 14 were shown previously [20]. At month 6, few people tested positive for neutralizing antibodies against Omicron (B.1.1.529), and no statistical difference in GMT or seropositivity rate was observed between the heterologous group and the homologous group, whether for the three-dose regimen or the two-dose regimen (Figure 3, Appendix A). The seropositivity rate differences between the two time points (day 14 and month 6) were 87.8% (95% CI: 76.7% to 99.0%) and 51.2% (95% CI: 33.1% to 69.3%) in the Convidecia boosting group and the CoronaVac boosting group of the three-dose regimen cohorts of participants aged 18–59 years and 75.3% (95% CI: 64.2% to 86.4%) and 31.9% (95% CI: 15.7% to 48.1%) for elderly participants. Compared with the GMTs of neutralizing antibodies against Omicron (B.1.1.529) at day 14, a greater than ninefold decline in the Convidecia boosting group in the two age groups was observed at month 6 (Appendix A).

### 3.4. The pseudovirus Neutralizing Antibodies against the Omicron (BA.4/5) Variant

The proportions of participants with seropositivity of pseudovirus neutralizing antibodies against the Omicron (BA.4/5) variant were 80% in the heterologous groups of the three-dose regimen, higher than those in the homologous groups (*p* < 0.001), and the seropositivity rates in the two-dose regimen groups were under 50.0% at day 14. For people over 60 years of age, no statistical difference was found, with seropositivity rates of 58.0–75.5% (Appendix A). At month 6, the percentage of participants tested positive for pseudovirus neutralizing antibodies to the Omicron (BA.4/5) variant was comparable among four groups in either of the two age groups (20.0–26.7% in 18–59 age groups, 41.7–65.0% in over 60 age groups) (Appendix A). The pseudovirus neutralizing antibody GMTs were 17.8–19.6 for people aged 18–59 years old and 25.9–44.4 for elderly participants (Figure 4).

Compared with day 14, percentages of participants with seropositivity reduced at month 6 in the heterologous groups. The seropositivity rate differences were 53.3% (95% CI: 28.5% to 69.6%) in the heterologous boosting group and 20.0% (95% CI: −12.3% to 47.5%) in the heterologous group for younger participants and 21.0% (95% CI: 16.3% to 38.5%) (heterologous boosting group) and 26.3% (95% CI: −1.3% to 49.0%) (heterologous group) for elderly participants (Appendix A).

The neutralizing antibodies against the SARS-CoV-2 prototype at month 6 were well correlated with RBD-specific IgG antibodies in the two age groups (r = 0.6–0.8, *p*-values < 0.001), but no significant correlations were shown between the neutralizing antibodies against the SARS-CoV-2 prototype and pseudovirus neutralizing antibodies against Omicron (BA.4/5), except in people in the heterologous prime-boost immunization group receiving two doses of CoronaVac plus Convidecia (r = 0.3, *p* = 0.024) (Appendix A).

## 4. Discussion

In this study, we reported the antibody responses induced by the heterologous boosting of Convidecia and homologous boosting of CoronaVac at month 6 in two age groups aged 18–59 years and 60 years or older who were primed with one dose or two doses of CoronaVac. Broadly speaking, the two age groups had similar levels of neutralizing antibodies against the SARS-CoV-2 prototype, the Delta variant, the Omicron (B.1.1.529) variant, and RBD-specific IgG antibody levels at month 6, and no correlations were observed between antibody levels and age. Only people who received three doses of CoronaVac presented statistically positive correlation of pseudovirus neutralizing antibodies against the Omicron (BA.4/5) variant and age.

At month 6, the heterologous regimen of CoronaVac plus Convidecia in the two age groups elicited higher GMTs of neutralizing antibodies against the SARS-CoV-2 prototype and RBD-specific IgG antibodies than the homologous regimen, whether in the three-dose regimen groups or in the two-dose regimen groups. As for neutralizing antibodies against the Delta variant, the higher GMTs and seropositivity rates of antibodies were also presented in the heterologous prime-boost immunization groups. The same results were shown in another antibody persistence study of heterologous prime-boost vaccination with CoronaVac plus orally aerosolized Ad5-nCoV [22], and the heterologous booster regimen with aerosolized Ad5-nCoV was more immunogenic and produced higher relative protection effectiveness compared to the homologous CoronaVac regimen [23]. However, due to low levels of neutralizing antibody titers against the Omicron (B.1.1.529) variant and pseudovirus neutralizing antibodies against the Omicron (BA.4/5) variant maintained at month 6, there was no significant difference between the heterologous and homologous immunization schedules in either age group.

Pseudovirus neutralizing antibodies against the prototype elicited by the three-dose regimen of mRNA-1273 have been reported over six times to decline at month 6 [17]. In our study, compared with the peak antibody levels at day 14, the GMTs of neutralizing antibodies against the SARS-CoV-2 prototype and RBD-specific IgG antibodies showed evident declines, with more than sixfold decreases at month 6 in the heterologous immunization schedule groups in the two age cohorts. The trend of antibody decline is described as the higher the antibody titer at day 14, the greater the decrease up to month 6. It is worth noting that the GMTs of neutralizing antibodies against Omicron (B.1.1.529) decreased about 10-fold in the two age groups only in participants receiving two doses of CoronaVac and one dose of Convidecia, and the rate differences were over 70% in heterologous boosting regimen groups and were 30–50% in homologous boosting groups. A 6.9-fold decrease in Omicron neutralizing antibodies was also shown for three doses of an mRNA vaccine in 4 months [24]. In addition, most of participants had negative neutralizing antibodies to the Omicron (B.1.1.529) variant at month 6. Similar results of poor performance in the persistence of neutralizing antibodies to the Omicron variant induced by prime-boost immunizations have also been reported [25,26,27]: the neutralizing response declined to levels below the detection limit of almost all individuals by 6 months, especially to those newer Omicron subvariants BA.2.75.2, BQ.1.1, and XBB.1.5 [28].

There are several limitations to this study. Firstly, participants in this study were from two different trials, and the miss rate at month 6 was high in the elderly cohort, which resulted in uncertainty in comparing the antibody durability between two age groups. Additionally, only a portion of people were sampled to detect the pseudovirus neutralizing antibodies against the Omicron (BA.4/5) variant—too small a sample size to observe significant difference between the different vaccination schedule groups or assess an evident decrease at month 6 compared with the peak level. Also, a wider range of cross-reactive antibodies were not detected, such as the neutralizing antibodies against the recent emerging variants of Omicron XBB and Omicron BF.7. However, our study reported and compared the 6-month antibody responses of heterologous and homologous prime-boost immunization based on two randomized controlled trials for people of all ages, which supplemented immunogenicity data of a COVID-19 vaccine booster dose.

## 5. Conclusions

Although the immunogenicity of Convidecia as a booster dose following two doses of CoronaVac performed better than a homologous prime-boost with three doses of CoronaVac, antibody durability of two prime-boost immunization schedules was observed to significantly decrease at month 6. Our conclusion supports the evidence to promote scientific prime-boost immunization of COVID-19 vaccines.

## Figures and Tables

**Figure 1 vaccines-11-01815-f001:**
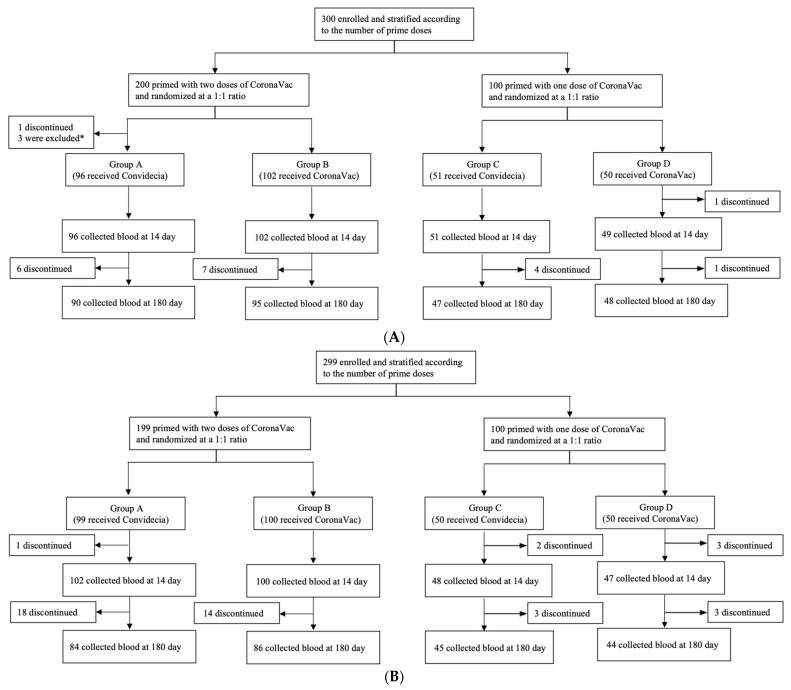
(**A**) CONSORT Flowchart for participants aged 18–59 years. Eighteen participants discontinued blood collection at day 180. * Two participants were randomized to group A but incorrectly received CoronaVac and were then classified into group B. Another participant was only primed with one dose but was incorrectly classified into group A (people primed with two doses). We reclassified this participant into group C. (**B**) CONSORT Flowchart for participants aged over 60 years. 34 participants discontinued blood collection at day 180.

**Figure 2 vaccines-11-01815-f002:**
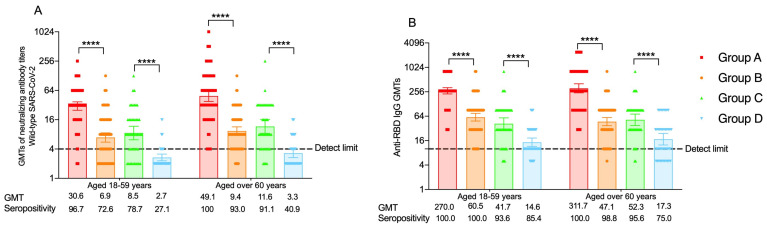
(**A**) The neutralizing antibodies against SARS-CoV-2 prototype at month 6 for people aged 18–59 years and over 60 years; (**B**) RBD-specific IgG antibodies at month 6 for people aged 18–59 years and over 60 years. Group A = participants who completed the two-dose primary series of CoronaVac and a booster dose of Convidecia. Group B = participants who completed the two-dose primary series of CoronaVac and a booster dose of CoronaVac. Group C = participants who received one dose of CoronaVac and another dose of Convidecia. Group D = participants who received one dose of CoronaVac and another dose of CoronaVac. GMT = geometric mean titer. **** *p* < 0.0001.

**Figure 3 vaccines-11-01815-f003:**
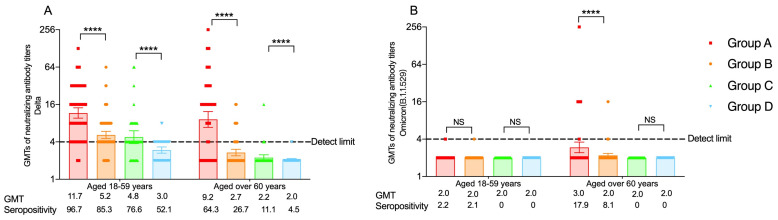
(**A**) The neutralizing antibodies against the Delta variants at month 6 for people aged 18–59 years and over 60 years; (**B**) The neutralizing antibodies against the Omicron(B.1.1.529) variants at month 6 for people aged 18–59 years and over 60 years. Group A = participants who completed the two-dose primary series of CoronaVac and a booster dose of Convidecia. Group B = participants who completed the two-dose primary series of CoronaVac and a booster dose of CoronaVac. Group C = participants who received one dose of CoronaVac and another dose of Convidecia. Group D = participants who received one dose of CoronaVac and another dose of CoronaVac. GMT = geometric mean titer. **** *p* < 0.0001; NS, not statistically significant.

**Figure 4 vaccines-11-01815-f004:**
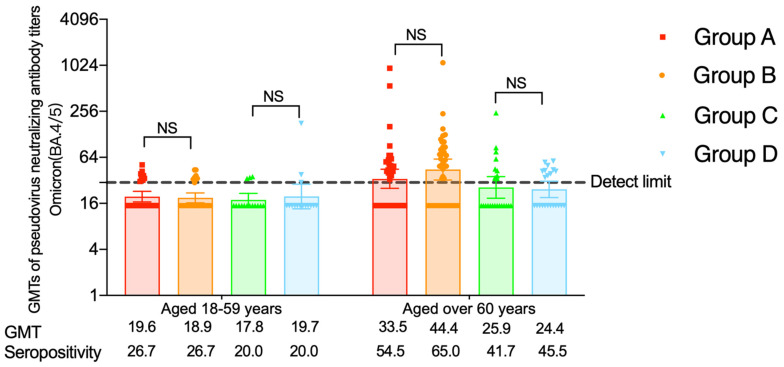
The pseudovirus neutralizing antibodies against the Omicron (BA.4/5) variant at month 6 for people aged 18–59 years and over 60 years. Group A = participants who completed the two-dose primary series of CoronaVac and a booster dose of Convidecia. Group B = participants who completed the two-dose primary series of CoronaVac and a booster dose of CoronaVac. Group C = participants who received one dose of CoronaVac and another dose of Convidecia. Group D = participants who received one dose of CoronaVac and another dose of CoronaVac. GMT = geometric mean titer; NS, not statistically significant.

**Table 1 vaccines-11-01815-t001:** Baseline characteristics of participants aged 18–59 years and aged over 60 years at month 6.

	Aged 18–59 Years			
	Group A (*n* = 90)	Group B (*n* = 95)	Group C (*n*= 47)	Group D (*n*= 48)
Age, years				
Mean (SD)	45.5 (9.2)	45.3 (9.0)	43.7 (9.8)	43.0 (9.0)
Median (IQR)	48.0 (43.0–51.0)	47.0 (41.0–52.0)	47.0 (36.0–51.0)	44.0 (38.5–49.5)
Sex (%)				
Male	56 (62.2)	59 (62.1)	26 (55.3)	28 (58.3)
Female	34 (37.8)	36 (34.9)	21 (44.7)	20 (41.7)
Time since the last priming dose of inactivated vaccine (months)				
Median (IQR)	3.2 (3.2–4.6)	3.8 (3.2–4.6)	1.8 (1.8–1.8)	1.8 (1.8–1.8)
	Aged over 60 years			
	Group A (*n* = 84)	Group B (*n* = 86)	Group C (*n* = 45)	Group D (*n* = 44)
Age, years				
Mean (SD)	66.9 (4.6)	66.7 (3.7)	70.2 (5.7)	69.6 (5.6)
Median (IQR)	66.0 (64.0–70.0)	66.0 (64.0–70.0)	71.0 (65.0–73.0)	70.5 (66.0–73.0)
Sex (%)				
Male	53 (63.1)	52 (60.5)	27 (60.0)	19 (43.2)
Female	31 (36.9)	34 (39.5)	18 (40.0)	25 (56.8)
Time since the last priming dose of inactivated vaccine (months)				
Median (IQR)	4.9 (4.7–5.0)	4.9 (4.7–5.0)	1.1 (1.1–1.2)	1.1 (1.1–1.1)

Data are *n* (%) or mean ± SD. or median (IQR). *n* = number of participants. % = proportion of participants. SD = standard deviation. IQR = interquartile range.

## Data Availability

The data presented in this study are available upon request from the corresponding author.

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
