# Peer review of "The 6-Month Antibody Durability of Heterologous Convidecia Plus CoronaVac and Homologous CoronaVac Immunizations in People Aged 18–59 Years and over 60 Years Based on Two Randomized Controlled Trials in China"

_vaccines, 2023, doi:10.3390/vaccines11121815_

Round 1

Reviewer 1 Report

Comments and Suggestions for Authors

In the current study, Jiang et al. aimed to understand the homologous vaccination regime (CoronaVac with a booster of CoronaVac) and the heterologous vaccination regime (CoronaVac with a booster dose of Convidecia) and their efficacy in neutralizing COVID-19 and its mutant variants in middle and old-age individuals. The primary objective was to enhance the efficiency of CoronaVac, a killed vaccine, by boosting it with a Non-Replicating Viral Vector, Convidecia. They evaluated the neutralizing efficiency against COVID-19 and its mutants in two age groups. The goal of this study is to enhance the effectiveness of the COVID-19 vaccination and to dispel myths surrounding the safety of heterologous vaccines. However, several aspects of the paper need clarification and refinement.

Major comments: Authors must address the major concerns that could enhance the quality of the manuscript.

1.    Why do the authors maintain a difference of 2-3 times the duration (in weeks) between the booster priming in the homogenous and heterogeneous groups compared to the last priming dose with the inactivated vaccine groups? Is there a specific reason for this approach? If so, kindly provide the rationale in the manuscript with appropriate references or explanations.

2.    In several instances, the sentences are excessively lengthy (Lines 196-201; 202-208), making it challenging to read and comprehend the results. To enhance understanding, it is necessary to break down these sentences and make them more concise.

3.    It would be more appropriate to divide the individuals into more than two age groups, yielding more suitable data than the current method and enhancing the understanding of the results.

Minor comments:  Authors are required to make the following minor changes for better understanding and clarification.

1.    Grammer and Typos: Several grammatical mistakes were there throughout the manuscript and must be rectified. For example, in line 26, where respectively is not necessary. A space has been omitted before the opening bracket in multiple locations throughout the manuscript.

2.    References should be appropriate and should not be misleading. For instance, in Reference 1, the authors discussed vaccination in various countries, but the reference they cited pertains to clinical trial data.

3.    In Line 80, mention the abbreviation (GMTs) when shown for the first time in the manuscript.

4.    In the materials section, it is mentioned that the groups were randomly divided based on the number of primary doses of CoronaVac, but in Fig. 1, the one dose of CoronaVac is missing.

5.    Line 124, after the sentence ends with a full stop.

6.    Line 150 insert space before 34 participants.

7.    Use a consistent abbreviation for all authors; however, some authors were mentioned by their full names.

Author Response

Major comments: Authors must address the major concerns that could enhance the quality of the manuscript.

  1. Why do the authors maintain a difference of 2-3 times the duration (in weeks) between the booster priming in the homogenous and heterogeneous groups compared to the last priming dose with the inactivated vaccine groups? Is there a specific reason for this approach? If so, kindly provide the rationale in the manuscript with appropriate references or explanations.

Thanks for the reviewer’s comment and it is a significant question to discuss. People were recommended to received a booster dose 6 months after completing the primary series of COVID-19 vaccines, so the interval time of 3-6 months between the last priming dose and the booster dose were designed in homogenous and heterogeneous prime-boost immunization groups in this study. For the other two groups, people only who received one-dose of CoronaVac did not complete the primary immunization, so the second dose was designed to vaccinate in about 1 month.

  1. In several instances, the sentences are excessively lengthy (Lines 196-201; 202-208), making it challenging to read and comprehend the results. To enhance understanding, it is necessary to break down these sentences and make them more concise.

We acknowledge the comment. We have broken down the sentences.

  1.  It would be more appropriate to divide the individuals into more than two age groups, yielding more suitable data than the current method and enhancing the understanding of the results.

Thanks for the reviewer’s comment. In our study, participants aged 18-59 years and over 60 years or old were two cohorts from two different heterologous immunization studies, and the safety and immunogenicity of the immunization have been reported respectively. Now we report the antibody durability for two age groups in this manuscript together. We also acknowledge that analysis among over two age groups might be more significant, but we have tried to conduct the correlation analysis between the age and antibody titers, and found that the correlation coefficient were near 0 in all the immunization procedures groups and no statistical significance were shown.

Minor comments:  Authors are required to make the following minor changes for better understanding and clarification.

1.Grammer and Typos: Several grammatical mistakes were there throughout the manuscript and must be rectified. For example, in line 26, where respectively is not necessary. A space has been omitted before the opening bracket in multiple locations throughout the manuscript.

Thanks for the reviewer’s correction. We have corrected these mistakes and add a space before the opening bracket.

  1. References should be appropriate and should not be misleading. For instance, in Reference 1, the authors discussed vaccination in various countries, but the reference they cited pertains to clinical trial data.

Thanks for the reviewer’s correction, and we have checked the references mentioned in this manuscript. But we think that the Reference 1 cited might be suitable.

  1. In Line 80, mention the abbreviation (GMTs) when shown for the first time in the manuscript.

Thanks for the reviewer’s remind, and we have added the abbreviation.

4.In the materials section, it is mentioned that the groups were randomly divided based on the number of primary doses of CoronaVac, but in Fig. 1, the one dose of CoronaVac is missing.

Thanks for the reviewer’s remind. It was our mistake to write ‘one’ to ‘two’. In the modified Figure, we have corrected it.

  1. Line 124, after the sentence ends with a full stop.

Thanks for the reviewer’s correction, and we have corrected.

  6.Line 150 insert space before 34 participants.

Thanks for the reviewer’s remind, and we have added a space.

7.Use a consistent abbreviation for all authors; however, some authors were mentioned by their full names.

Thanks for the reviewer’s comment, and we have checked.

Reviewer 2 Report

Comments and Suggestions for Authors

Thank you for sharing your manuscript investigating the AB duration of heterologous and homologous CoronaVac immunization strategies among adult participants. Please find some comments that could help to improve your article.

L52: Please include some brief description of CoronaVac in your manuscript for readers not familiar with this vaccine. L77: The same applies to Convidecia. 

L79: Please stated in your manuscript how the health status was assessed/confirmed. L95: The same applies here. 

L77-80: Please revise this sentence for more clarity.

L98-100: What testing did you perform to assess the exclusion criteria stated?

L101: How did you perform the random assignment of participants into groups A-D? Please state in your manuscript.

L106: Please define "adverse reactions" within the scope of your research performed. 

General comment: The section 2.2 Outcomes seems like a mixture of expected endpoint and laboratory investigation of neutralizing antibodies. Please revise this section for more clarity and include more details on the assay stated. 

L127: As you are aiming to statistically sound data, what is the sample size you targeted? 

L132: A reader may not know your protocol, thus provide details on the serum collection.

L140: If you truly calculated the median time interval between the doses administered, how comes you stated "about 3-4 months"? L140-143: Please be consistent when reporting figures, i.e., on the one hand you state "about" and on the other hand you state an exact figure with one decimal place. 

L143: I think the sentence "more detailed ....table 1." does not fit here and needs to be stated earlier. 

L161: What unit does accompany 150.3? The same for L166, 167 and the forth. Please also address this for Figure 2, 3 and 4. Also for Figure 4, it is not clear which bar resembles which group; please revise accordingly. 

Comments on the Quality of English Language

Please see above. 

Author Response

1.L52: Please include some brief description of CoronaVac in your manuscript for readers not familiar with this vaccine. L77: The same applies to Convidecia. 

Thanks for the reviewer’s remind, and we have added some brief description of CoronaVac and Convidecia.

2.L79: Please stated in your manuscript how the health status was assessed/confirmed. L95: The same applies here. 

Thanks for the reviewer’s comment. We have added the description of tests for the inclusion and exclusion criteria, and those who were satisfied with the inclusion criteria but unsatisfied with the exclusion criteria were defined as the health status.

3.L77-80: Please revise this sentence for more clarity.

Thanks for the reviewer’s comment, and we have modified.

4.L98-100: What testing did you perform to assess the exclusion criteria stated?

Thanks for the reviewer’s comment. We have added the description of tests for the exclusion criteria.

5.L101: How did you perform the random assignment of participants into groups A-D? Please state in your manuscript.

Thanks for the reviewer’s comment. We have added the description of randomization.

6.L106: Please define "adverse reactions" within the scope of your research performed.

Thanks for the reviewer’s comment. The primary endpoint of incidence of adverse reactions within 28 days after vaccination as the primary endpoint included solicited injection site events(pain, redness, swelling, induration, itch and cellulitis) and systemic events(fever, malaise, muscle ache, joint pain, fatigue, nausea, headache), which had been reported. In this manuscript, we just reported the antibody durability and safety results were not concerned.

General comment: The section 2.2 Outcomes seems like a mixture of expected endpoint and laboratory investigation of neutralizing antibodies. Please revise this section for more clarity and include more details on the assay stated. 

Thanks for the reviewer’s comment. We have revised Methods Part and added more details on the assays.

L127: As you are aiming to statistically sound data, what is the sample size you targeted? 

Thanks for the reviewer’s comment. We have added the description of sample size in the Methods.

L132: A reader may not know your protocol, thus provide details on the serum collection.

Thanks for the reviewer’s comment. We have added the description of serum collection.

L140: If you truly calculated the median time interval between the doses administered, how comes you stated "about 3-4 months"? L140-143: Please be consistent when reporting figures, i.e., on the one hand you state "about" and on the other hand you state an exact figure with one decimal place. 

Thanks for the reviewer’s comment. We revised the description of median time interval between the doses in the manuscript.

L143: I think the sentence "more detailed ....table 1." does not fit here and needs to be stated earlier. 

Thanks for the reviewer’s comment. We also added the label of Table 1 in the front.

L161: What unit does accompany 150.3? The same for L166, 167 and the forth. Please also address this for Figure 2, 3 and 4. Also for Figure 4, it is not clear which bar resembles which group; please revise accordingly. 

Thanks for the reviewer’s comment. 150.3 was the peaked GMT of neutralizing antibodies against SARS-CoV-2 prototype in 18-59 years group at day 14, which has been published(Nat Med. 2022 Feb;28(2):401-409. doi: 10.1038/s41591-021-01677-z.). Also, we have revised the Figure 4. As the number of line changed, I am sorry I can not understand the questions for L166, L167, Figure 2, 3 and 4.

Reviewer 3 Report

Comments and Suggestions for Authors

This is an interesting study, but methodology isn't always described very clearly, so my comments are directly in the PDF.

Author Response

1.I find the title unclear, it should be rephrase  so that the purpose of this study is understood. the study design and the country where the study took place should also be added.

Thanks for the reviewer’s comment. We have modified the title.

2.References are not in the right format for this journal.

Thanks for the reviewer’s comment. Instructions for Authors of Vaccines recommend ACS style guide as this manuscript used.

3.I find introduction a little short. More information is needed on the background to the study, so that we can better understand its interest and stakes.

Thanks for the reviewer’s comment. We have added some background in the Introduction.

4.Even though these 2 RCTs have been described elsewhere, I think it's important to give to the readers some information on their development, population and sample size.

Thanks for the reviewer’s comment.We have added the description about the participants inclusion and exclusion criteria, randomization, serologic assays and sample size.

5.This part is very hard to read, it should be summarized in sub-chapters so as not to lose the reader.

Thanks for the reviewer’s suggestion, and we have added the sub-chapters.

6.I don't quite understand, if most of the results are already present in studies 18 and 19, what's the point of this study?

Thanks for the reviewer’s comment. Studies 18 and 19 mainly reported the immunogenicity and safety of heterologous immunization with Ad5-nCOV at day 14 and day 28. In this study, we principally reported the 6-month antibody durability of heterologous Convidecia plus CoronaVac and homologous CoronaVac immunization. We presented the previous results here in order to compare the peak of antibody level to the month 6.

7.I find the section on interpreting results and comparing them with other studies a little weak. It should be expanded to highlight the novelty of the results found by the authors.

Thanks for the reviewer’s comment. We have added the discussions about comparing our results with other studies, and highlighted the novelty of our works.

Round 2

Reviewer 1 Report

Comments and Suggestions for Authors

The authors have addressed all the concerns in the revised manuscript. 

Reviewer 2 Report

Comments and Suggestions for Authors

Thank you for sharing the revised manuscript and for addressing all my comments.

Reviewer 3 Report

Comments and Suggestions for Authors

thanks to the authors for their additions and corrections, this article seems acceptable in this form.